# Potential Role of Neutrophil Extracellular Traps in Cardio-Oncology

**DOI:** 10.3390/ijms23073573

**Published:** 2022-03-25

**Authors:** Kai-Hung Cheng, Gregory P. Contreras, Ting-Yu Yeh

**Affiliations:** 1Division of Cardiology, Department of Internal Medicine, E-Da Cancer Hospital, Kaohsiung 82445, Taiwan; kaihung0218@gmail.com; 2College of Medicine, I-Shou University, Kaohsiung 82445, Taiwan; 3Auxergen Inc., Columbus Center, 701 East Pratt Street, Baltimore, MD 21202, USA; greg@auxergen.com

**Keywords:** neutrophil extracellular traps (NETs), cardio-oncology (CO), cancer therapeutics-related cardiovascular dysfunction (CTRCD), cancer treatment-related cardiovascular toxicity (CTRCT)

## Abstract

Cardiovascular toxicity has emerged as the leading cause of death in patients undergoing cancer treatment. Thus, cardio-oncology (CO) care must also focus on the prevention and management of related cardiovascular (CV) complications caused by cancer therapy. Neutrophil extracellular traps (NETs)—entities with released DNA, proteases, proinflammatory and prooxidative substances from blasted neutrophils—play an important role in cancer proliferation, propagation metastasis, and incident CV events (acute coronary syndrome, thromboembolic events, and heart failure). Although NETs have been shown to be involved in cancer progression and incident CV events, little is known about their relationship with cardio-oncology, especially on cancer treatment-related cardiovascular toxicity (CTRCT). This review aims to explore the evidence of the impact of NETs on cancer, CV events, and CTRCT, and the possible solutions based on the mechanism of NETs activation and NETs released toxic substances.

## 1. Introduction of Cardio-Oncology

New advances in cancer therapies have increased the survival rate of cancer patients tremendously [1]. However, an emerging issue of these cancer therapies is the associated side effects on the cardiovascular (CV) system which lead to different spectrums of morbidity and mortality [2]. Recently, cancer treatments have been expanded from cytotoxic chemotherapy, radiation therapy and surgery, to targeted and immune-based therapies [3]. Cardiotoxicity refers to the direct harmful effects of cancer treatments on the CV system and/or the acceleration of CV diseases (CVDs) in addition to traditional CV risk factors [2,4]. The origins of cardio-oncology (CO) can be traced back to 1 July 2000, when the MD Anderson Cancer Center initiated a comprehensive program to diagnose, treat and manage all CVDs of cancer survivors. The main focus of CO research is on the prevention and management of related CV complications caused by cancer therapies.

Recent studies have supported the notion that recurrent dysfunction with rechallenge from chemotherapy treatment may lead to intractable heart failure or death without suitable interventions and adjustments in the chemotherapy regimen. The concept of type I irreversible and type II reversible cardiotoxicity was introduced by Ewer and Lippman, whose studies mainly focused on anthracyclines and trastuzumab [5,6]. Emerging evidence has been reported that doxorubicin, an anthracycline that blocks topoisomerase 2, can cause type I cancer therapeutics-related cardiovascular dysfunction (CTRCD). It has been observed that type I CTRCD is dose-dependent, cumulative, progressive, and irreversible. Multiple signatures of myocyte damage by doxorubicin (vacuolar swelling, myofibrillar disarray, and cell death, etc.) have been discovered in myocardial biopsies under electron microscopy [7]. Alternatively, type II CTRCD is considered reversible, because it is relatively safe to rechallenge, with a high likelihood of near recovery in 2–4 months after interruption [7]. Trastuzumab, a monoclonal antibody for cancers with large amounts of human epidermal growth factor receptor 2 (HER2), provides the best example of type II CTRCD where the myocardial damage is not considered to be cumulative, dose-dependent, or progressive. Electron micrographic observations of myocardial biopsies have shown that trastuzumab does not result in cell damage [7]. Type II CTRCD has also been observed in other cancer treatments, such as anti-HER2-targeted therapies (the monoclonal antibodies pertuzumab and trastuzumab emtansine), and the tyrosine kinase inhibitor (TKI) lapatinib [6,8].

Recently, studies have raised questions about the classification of type I and II CTRCD, as well as the question of whether type I CTRCD is irreversible [9,10], while type II CTRCD is reversible [11]. To effectively evaluate different patterns of cardiotoxicities, scores of anonymized profiles have to be combined and reviewed, including drug treatments (the type, timing, duration, and combination) and the patient’s genetics and comorbidities.

It is well known that radiotherapy may cause damage of the pericardium, coronary arteries, valves, endocardium, and myocardium, and that these symptoms may occur in the acute (<6 months) or late phase (3–30 years) [12,13]. For example, it has been reported that breast cancer patients with radiotherapy have a greater risk of coronary heart disease and cardiac death (30% and 38%, respectively) compared to patients without radiotherapy [14]. Patients receiving radiotherapy together with anthracyclines also have higher CV risk [15,16], and patients with left-sided breast cancer have higher risk of heart injury (1.4-fold) than those with right-sided breast cancer [17].

Radiation volume and dose are direct risk factors of radiotherapy to the heart and its substructures in coronary arterial disease [18,19]. Darby et al. have reported that, in 2168 women who underwent radiotherapy for breast cancer between 1958 and 2001 in Sweden and Denmark, the rate of major adverse cardiac events (i.e., coronary revascularization, myocardial infarction, or CV death) increased linearly by 7.4% per Gray (Gy) increase in mean heart dose [20]. To reduce cardiac exposure to radiation, several procedures of risk mitigation have been introduced. These procedures include proton irradiation, custom blocks for the heart, displacement maneuvers such as prone positioning and deep inspiratory breath holding, intensity-modulated techniques, and intraoperative irradiation and/or brachytherapy [18,21,22].

CTRCD is defined as a reduction in global longitudinal strain (GLS) of >15% from the baseline, and the left ventricle ejection fraction <50% is generally considered abnormal and an early sign of left ventricular subclinical dysfunction [2], as well as an early indicator of heart failure. In addition to GLS [23,24], the high-sensitivity cardiac troponins (hsTn) are also predictive markers of heart failure [2,25,26]. However, it is worth noting that not only CTRCD but also the entire CV system can be affected by cancer treatments. Such abnormal occurrences in the CV system include hypertension, endothelial and vascular dysfunction, accelerated atherosclerosis, thrombosis and bleeding, pulmonary hypertension, pericardial disease, QT prolongation, and conduction disease/arrhythmias [4,27,28,29,30,31,32]. Thus, cancer treatment-related cardiovascular toxicity (CTRCT) is defined as cancer therapy-related complications in the entire CV system. Different anti-cancer therapies (chemotherapy, targeted therapy, hormone therapy, immunotherapy, radiation therapy, and surgery) and bone marrow transplantation have their own relevant CV concerns.

## 2. Neutrophil Extracellular Traps (NETs): Introduction, Mechanism of Toxicity Focusing on Cardiovascular Toxicity and Contribution to the Cardiovascular Events

### 2.1. Introduction of NETs: Formation, Definition, and Components

In addition to dendritic cells and macrophages, neutrophil acts as a phagocyte to control invasive pathogens, and is an essential component of innate immunity [33]. Under some specific stimuli, such as phorbol 12-myristate 13-acetate (PMA) or infection, monocytes can differentiate to become macrophages and dendrite cells [34], but neutrophils form neutrophil extracellular traps (NETs) [35]. In addition to phagocytosis, neutrophils have another more powerful antimicrobial mechanism: the release of NETs.

Neutrophils are able to respond to a variety of stimuli, including various pathogens, such as bacteria (including its wall components—lipopolysaccharides), fungi, protozoa, viruses, antibodies and immune complexes, cytokines and chemokines (IL-8, TNF), microcrystals, and other pharmacological stimuli, as well as PMA (mimic the action of diacylglycerol activating protein kinase C), calcium (ionomycin, A23187), and potassium (nigericin) ionophores in vitro experiments. Neutrophils undergo a specialized process of programmed death (which does not require the activation of caspases as apoptosis) resulting in chromatin decondensation and subsequent NETs formation. NETs were first shown to kill extracellular pathogens in order to minimize damage to host cells in 2004 [36,37]. 

The most striking characteristics of NETs are DNA strands bounded with pro-interleukin-1α (pro- IL-1α) which can activate endothelial cells and produce more inflammatory cytokines. NETs are also composed of many protein/enzyme components and chemicals, such as myeloperoxidase, neutrophil elastase, tissue factors that can activate Factor VII to initiate thrombosis, and highly prooxidant species hypochlorous acid (HOCl) [38]. Myeloperoxidase (MPO) is a peroxidase which takes heme as a cofactor and produces hypochlorous acid (HOCl) from hydrogen peroxide (H_2_O_2_) and chloride anion (Cl^−^) during the neutrophil’s oxidative burst and takes H_2_O_2_ as an oxidizing agent to oxidize tyrosine to tyrosyl radical [39]. The HOCl and tyrosyl radical are both cytotoxic and are used to kill pathogens by the neutrophil as a part of innate immune response. However, both HOCl and tyrosyl radicals can cause oxidative and inflammatory damages to human tissues. Neutrophil elastase hydrolyzes proteins within specialized neutrophil lysosomes, known as azurophil granules, and degrades both the outer membrane protein A and the virulence factors of bacteria [40]. Neutrophil elastase can also disintegrate extracellular matrices in human tissues, such as collagen-IV and elastin, following NETosis (the process of the formation of NETs). Thus, the formation of NETs is presumed to kill pathogens by oxidation through MPO and HOCl, and to cleave laminin barriers with neutrophil elastase in order to kill pathogens far afield from vessels. Because this powerful anti-microbial action is non-specific, dysregulated NETosis has been reported to be associated with acute and chronic inflammatory disease, tissue damage, organ fibrosis, hypercoagulability, thrombosis, atherosclerosis and cancer metastasis, and the damage associated with COVID-19 infection [38,41,42,43]. NET formation results in local thrombosis to prevent pathogens from spreading. In addition, NETs have been also implicated in several organ fibrosis, including pulmonary fibrosis. In the alveolar and interstitial lung space of mice treated with bleomycin, Suzuki et al., have shown that lung fibrosis was induced by NETs and dependent on peptidyl arginine deiminase 4 (PAD4) [44]. Although NETs have been shown to be involved in cancer progression and incident CV events, little is known about their relationship with CO. This review will focus on the potential role of NETs in CO (Summary Figure 1).

### 2.2. Mechanism of Toxicity Focusing on Cardiovascular Toxicity and Contribution to Cardiovascular Events

NETs are reported to be present in human atherosclerotic vessels [45] and coronary specimens after acute myocardial infarction. NETs burden is correlated positively with infarct size, and negatively with ST-segment resolution and all markers of NETosis. Markers including nucleosomes, double-strand DNA, neutrophil elastase, and MPO, have been found to be elevated in subject lesions [46,47]. Professor Peter Libby has described some links between NETs and acute coronary syndrome with plaque erosion (first identified by Professor Ik-Kyung Jang). NET-associated cytotoxicity is observed during plaque erosion when NETs, released at sites of disturbed flow, induce endothelial cell desquamation [48]. This process involves disturbed flow-related endothelial cell death, detachment, and basement membrane breakdown through matrix metalloproteinases in response to some specific growth factors, platelet, and neutrophil activations followed by NETs formation. NETs’ elaborate strands of nuclear DNA (with proteins including MPO, neutrophil elastase, and tissue factor) initiate coagulation cascades. A pro-oxidant HOCl, as well as IL-1α formed from Pro-IL-1α by the local neutrophil serine proteinase cathepsin G, initiates and propels the progression of inflammation. NETs, hyaluronan fragments, and damage-associated molecular partners further amplify and propagate the endothelial activation and apoptosis, and inflammation and thrombosis, in a vicious cycle [38,49]. One study reported that C-reactive protein induces a concentration-dependent NET synthesis in patients with heart failure [50]. A link has been proposed between NETs-related coronary microvascular and macrovascular thrombosis through some possible mechanisms, involving mitochondrial DNA, high-mobility group box 1, fibronectin extra domain A, and galectin-3 and pattern recognition receptors, leading to the increase of heart failure [51,52,53,54].

## 3. NETs in Cancer Itself

One of the common complications and major causes of mortality in cancer patients is cancer-associated thrombosis. It has been reported that cancer patients have a higher risk of initial venous thromboembolism (VTE) (4–7-fold), recurrent VTE (3-fold), anticoagulation-associated bleeding (2-fold), death from VTE (10-fold), and arterial thromboembolism (2-fold) compared to those without cancers [55,56]. Risk factors in cancer-associated thrombosis can be patient-related (e.g., ethnicity, age, comorbidities, etc.), cancer-related (e.g., histology, grade, primary site, and initial period after diagnosis, etc.), treatment-related (e.g., chemotherapy, surgery/hospitalization, antiangiogenics, central venous cannulation, erythropoietin stimulating agent/transfusion-related, etc.), and are also related to important biomarkers (e.g., leukocyte and platelet count, D-dimer, hemoglobin, etc.) [57]. Recent reports have revealed that NETs provide skeletal support for procoagulant molecules in association with red blood cells and platelets to promote thrombosis in vitro and in vivo. NETs contain complicated protein components, which can activate the endogenous coagulation pathway promoting thrombosis. Therefore, NETs appear critical to the formation of cancer-related arterial and venous thrombosis [58,59].

Currently, the roles of NETs in the initiation, formation, and metastases of cancer and in cancer-associated thrombosis are being extensively investigated. It has been reported that cancers can induce the formation of NETs partly due to their (1) hypoxic microenvironment, (2) higher expression of transcription factor HIF-1α, (3) higher reactive oxygen species under high oxidative stress, and (4) cancer-secreted cytokines, proteases, and exosomes [60]. Hypoxic microenvironments and oxidative stresses which promote TGF-β1 secretion are key inducers of endothelial-to-mesenchymal transition (EMT). TGF-β plays an essential role in the EMT process in the tumor microenvironment during tumorigenesis and metastases. EMT is also an important contributor to the microenvironmental plasticity of cancerous tumors. Endothelial cells undergoing tumor-mediated EMT show functional alterations. This includes greater migratory capacity, higher proliferation rates, and a gain of contractile capacity and loss of their angiogenic ability to form capillary-like tubes [61]. NETs were shown to drive EMT to facilitate tumorigenesis [62,63,64], growth [64], and metastasis [64,65]. Therefore, NETs can promote tumor growth and progression while accumulating in the tumor microenvironment [66]. Because neutrophils are also responsible for T cell exclusion and hypoxia, NETs also foster tumor spread far afield by shielding cancer cells and thwarting the attack of circulating cytotoxic lymphocytes [67]. NETs-related neutrophil elastase, matrix metalloproteinase 9, and cathepsin G break the intercellular junction and basement membrane by proteolysis of VE-cadherin, thus increasing endothelial cell permeability. This process also activates endothelial cells to recruit circulating tumor cells which are captured by NETs to promote the cancer metastasis [62]. NETs’ link to cancer-associated thrombosis is related to the induced thrombosis cascade by the exposed tissue factors [68,69,70].

## 4. Effects of Cancer Treatments on Neutrophil Amount, Accumulation in Tissue, Activation, and Possible Effects of NET Formation

### 4.1. Type I CTRCD Agents

Khan et al. showed that anthracyclines (e.g., epirubicin, daunorubicin, doxorubicin, and idarubicin) consistently suppress both NADPH oxidase-dependent and -independent types of NETosis in human neutrophils, ex vivo [71]. Doxorubicin is well known to cause type I CTRCD [7]. Although study into anthracycline-induced cardiotoxicity has been focused on the effects on cardiomyocytes that lead to contractile dysfunction, the neutrophil recruitment in addition to endothelial injury is increasingly recognized as a likely mechanism for cardiotoxicity [72]. In patients with breast cancer receiving doxorubicin-based chemotherapy, the high plasma levels of NETs were associated with doxorubicin-related cardiotoxicity [73]. Other type I CTRCD chemotherapy drugs include alkylating agents (e.g., busulfan, carboplatin, carmustine, chlormethine, cisplatin, cyclophosphamide, and mitomycin); topoisomerase inhibitors (e.g., etoposide); tretinoin; vinca alkaloids; and antimetabolites (e.g., cladribine, cytarabine, and 5-FU) [7]. The alkylating agents such as carboplatin and cisplatin, and the vinca alkaloids such as vincristine, have shown to increase reactive oxygen species (ROS) formation, which leads to inflammation and neutrophil-mediated cell injury [74]. 5-FU, one of the antimetabolites, has also been shown to increase the amount of NETs [75].

### 4.2. Type II CTRCD Agents

Trastuzumab, lapatinib, pertuzumab, and trastuzumab emtansine remarkably extend HER2 breast cancer survival. These drugs are known to cause type II CTRCD [7].

In a case report of life-threatening trastuzumab-induced interstitial lung disease (pulmonary capillaritis), circulating NETs are correlated with the clinical severity. Chang et al. reported that MPO-DNA complex is a useful biomarker to monitor vasculitis activity [76]. However, little is known about the circulating NETs level vis-à-vis cardiotoxicity in these anti-HER2 agents. As to TKI, one study showed that ponatinib augmented the NETs increase, which remains a possible cause of vascular toxicity [77].

### 4.3. Radiotherapy

Tumor-associated neutrophils are known to have antitumor and protumor phenotypes, depending on the tumor microenvironment. Radiotherapy can activate neutrophil recruitment. One previous study showed that the combination of radiotherapy and granulocyte colony-stimulating factor (G-CSF) induces the polarization and ROS production of anti-tumor neutrophils that triggers apoptosis of tumor cells [78].

On the other hand, after radiotherapy, NETs can facilitate tumor progression by recruiting polymorphonuclear neutrophils (PMNs) to the tumor immune microenvironment. Liu et al. reported that the accumulation of lymphocytes, particularly CD8 cytotoxic T cells, was enhanced by G-CSF [79]. ROS accumulation induces mesenchymal-epithelial transition and plays a major role in this process. In one observational study in patients with bladder tumors, NETs were responsible for poor response to radiotherapy, and caused persistent disease post-radiotherapy, including a high tumoral PMN-to-CD8 ratio, which is associated with worse overall survival [80]. Muravlyova et al. reported that colorectal cancer patients (Stage II and III) are more likely to have NETs in the center and on the periphery of the tumor, and in healthy tissues adjacent to the tumor, than patients of Stage I [81]. Radiation therapy as a pre-operative preparation also contributed to an increase in the number of NETs in the center of the tumor in patients with Stage I rectal cancer.

### 4.4. Immunotherapies

Programmed cell death protein-1 (PD-1), a key immune checkpoint receptor, mediates immunosuppression and contributes to the limited clinical efficacy of chimeric antigen receptor T (CAR-T) cells in solid tumors [82]. CAR-T cell therapy combined with anti-PD-1 antibody treatment has shown encouraging antitumor activity in patients [83,84]. In a murine patient-derived xenograft model of early-stage non-small cell lung carcinoma devoid of host lymphoid cells, Martin-Ruiz et al. observed that the antitumor effect of anti-PD-1 treatment is likely due to a mechanism independent of T lymphocytes, suggesting that neutrophils may act as PD-1 inhibitor effector cells responsible for tumor regression by necrotic extension [85]. 

CAR-T therapy-associated cytokine release syndrome and immune checkpoint inhibitor (ICI)-associated myocarditis have become the major focus of immunotherapy-related cardiotoxicities [86]. It is known that ICI-related cardiotoxicity is uncommon but usually fatal. According to the literature, ICI-related myocarditis is reported with low incidence (0.04–1.14%) but is associated with a high mortality rate (25–50%), and it appears to onset early after the initiation of therapy [27,87,88,89,90]. It has been reported that combination ICI therapy significantly increased the risk of myocarditis from 0.06% to 0.27% [27,88]. One review of 101 fatal cases revealed that 64% occurred after the first or second ICI dose [89], and another clinical study reported that some fatal cases occurred after the first ICI dose [88].

So far other risk factors of ICI-related myocarditis remain not completely clear, while several studies have suggested underlying autoimmune diseases, diabetes mellitus, and pre-existing CVD may be associated with NET formation [27,90,91,92]. Elevated troponin and abnormal ECG findings are common in the diagnosis of myocarditis [27,93]. Cardiac magnetic resonance with late gadolinium enhancement may be useful for an early diagnosis of myocardial edema, which is present in less than 50% of those with ICI-associated myocarditis [93,94]. The gold standard for diagnosis of myocarditis is the endomyocardial biopsy. However, because this procedure is invasive with the risk of complications, endomyocardial biopsy is underutilized; furthermore, many hospitals do not have that expertise [93,95]. In a recent study with patients with acute ischemic stroke [96] and a study of COVID-19 [97], it has been suggested the neutrophil-lymphocyte ratio (NLR) in peripheral blood and the expression of NETs within stroke emboli is well correlated. According to a study by Naik et al., a low NLR, which also indicates a low grade of NETs tumor microenvironment, is associated with a better long-term survival to anti–PD-1-based immune checkpoint inhibitors in melanoma patients with brain metastases previously naive to anti-PD-1 therapy [98]. In Moey et al.’s study of 196 ICI-treated lung cancer patients, NLR and C-reactive protein were significantly elevated at the time of major adverse cardiac events (MACE, including myocarditis, non-ST-segment elevated myocardial infarction, supraventricular tachycardia, and pericardial disorders) compared to baseline values in ICI-treated patients [99].

It has been shown that myocarditis cardiotoxicity is related to activated T cells [100], and less than 50% of patients have been reported to respond to high doses of immunosuppressants or corticosteroids [27,101]. NETs molecules have been proposed as mechanisms of immunotherapy resistance and potential targets for emerging candidate drugs [102]. It has been further suggested that a high number of tumor-associated neutrophils in the tumor microenvironment, as well as aberrant NETs release, indicate poor response to immunotherapies for several cancers. Similarly, cancer patients with high circulating IL-8 appear to predict poor responses from checkpoint-based immunotherapy [102]. It is suggested that IL-8 may facilitate cancer metastasis and escape from cytotoxic immune cells because IL-8 not only recruits neutrophils in the tumor microenvironment but also triggers NET formation. Therefore, it has been proposed that IL-8 can awaken micro-metastasis through direct effects on cancer cells, as well as from changes in the tumor microenvironment [103] (Figure 2).

## 5. Possible Treatment Modalities

In chemotherapy patients with breast cancer, Todorova et al. have showed that nucleosomes (a surrogate marker of NETs) and thrombin–antithrombin complex (a surrogate marker of high thrombin level) were significantly increased in patients with cardiotoxicity (defined as reduction of left ventricle ejection fraction >10% from baseline in this study) after the first dose of doxorubicin-based chemotherapy [46].

Chronic radiation exposure, like radiotherapy, is found to propagate atherosclerosis by stimulating neutrophil infiltration [104]. In addition, NETs in the microenvironment are also involved in tumor radio-resistance [80]. Therefore, mitigating the burden of NETs may be beneficial for stopping cancer progression, improving response for cancer treatment, as well as preventing CTRCT. Since the potential benefits are great in both the oncology and cardiology fields, new studies involving both fields are emerging.

There are several possible candidates for mitigating NETs burden, including potent anti-platelet therapy to inhibit platelet activation, deoxyribonuclease (DNase I) to break down circulating free DNA, the inhibition of peptidyl arginine deiminase-4 (PAD4) to form NETs, and the inhibition of MPO and anti-cytokine therapy (e.g., targeting IL-1 isoforms, IL-8, or IL-6) to alleviate oxidative stress and inflammation from NETs [38]. In addition, serine proteases inhibition is also reported to reduce the burden of NETs [105]; however, some serine proteases are reportedly involved in cancer proliferation and metastasis [106,107] (Summary Figure 1).

Despite the clinical importance of NETs in a variety of diseases, including lung injury and cancer, so far to date the only FDA-approved NET-targeting drug is the inhaled drug dornase alfa (Genentech’s Pulmozyme for recombinant DNase I). It has been shown that disrupting existing NETs by DNase I can enhance the therapeutic efficacy of tumor immunotherapy and attenuate metastatic spread [108]. Studies have shown that DNase I plays an important role in the degradation of NETs, which consequently dampens the resistance to anti-PD-1 blockade by attenuating tumor growth in a mouse colorectal cancer model [109]. According to this model, DNase I can decrease tumor-associated neutrophils as well as the formation of MC38 tumor cell-induced NET formation in vivo. The authors suggest that the inhibition of NETs by DNase I treatment leads to the reversal of anti-PD-1 blockade resistance through increasing CD8+ T cell infiltration and cytotoxicity [109] (Figure 2). However, it is worth mentioning that the systemic administration of DNase I could raise safety issues, potentially impairing host immune defense against a variety of infections [108]. One animal study revealed that that PAD4 might be another therapeutic target, as PAD4 deficiency can ameliorate the formation of NETs and subsequent lung fibrosis [44].

## 6. Unsolved Questions

Since very few studies have focused on NETs in CO, there are many areas of investigation warranted in this field. First, although the surrogate markers of NETs were significantly increased in patients with cardiotoxicity after the first dose of doxorubicin-based chemotherapy in patients with breast cancer [46], it is still unclear if NET (presence/absence) can predict CTRCD better than current recommended echocardiography-based GLS [23,24] and high-sensitivity cardiac troponin (hsTn) [2,25,26]. Perhaps the combination of all three (NETs + GLS + hsTn) can have the best predictive power. The Khorana score, introduced in 2008, is now the standard to predict cancer venous thromboembolism (VTE) which is composed of five clinical and pre-chemotherapy laboratory parameters: (1) primary tumor site (+1 or 2 points), (2) platelet count of 350 × 10^9^/L or more (+1 point), (3) hemoglobin concentration of 100 g/L or lower or use of erythropoiesis-stimulating agents (+1 point), (4) leukocyte count of 11 × 10^9^/L or higher (+1 point), and (5) a Body Mass Index of 35 kg/m^2^ or higher (+1 point) [110]. It would be interesting to determine whether NETs can offer better predictions for overall and individual thrombosis events including VTE, acute coronary syndrome, and stroke than the Khorana score. Therefore, it is important to initiate a nationwide or an international multicenter prospective randomized double-blinded study to evaluate the impact of NETs on CTRCD and thrombosis events to determine the causalities of NETs on CTRCD and thrombosis in patients with cancer. Second, there has been little discussion about NETs in association with other cells of innate immunity, especially for dendrite cells and macrophages. One animal study revealed that NETs could promote macrophage-related inflammation and accelerate atherosclerosis [111], but it is not clear it is an association or a causality nor are the signaling pathways among them apparent. Third, it is also important to know the association between NETs and modified low-density lipoprotein (mLDL) in atherosclerosis after cancer treatment. High circulating mLDL (such as lipoprotein(a) [Lp(a)] and electronegative LDL, especially for L5, …etc. through oxidation, glycosylation, carbamylation, and glycoxidation) is a well-known risk marker for CVDs and has been shown experimentally to be proatherogenic both in vitro and in vivo [112]. One review mentioned that mLDL could induce NETs formation, but it is not clear if NETs participate the formation of mLDL through their MPO and other proteases [113]. Additionally, it would be interesting to know if NETs can accelerate and participate in the formation of atherosclerosis in the long term. Fourth, it is also unclear if NETs are the keys to both cancer metastasis and CTRCT, or if can we resolve both difficult problems by clearing the NETs. Therefore, future studies should explore these questions, which are crucial for cardio-oncologists, cardiologists, and oncologists (Table 1).

## 7. Perspective

NETs play important roles in cancer progression and metastasis, incident acute coronary syndrome, thromboembolic events, and cancer treatment-related CTRCT. The mitigation of NETs in the CO field is an emerging issue. Studies of the mechanisms tackling platelet activation and alleviating NETs-related thrombosis and inflammation cascades, as well as anti-oxidative stress, have important questions to answer. However, there are still many unmet needs concerning NETs in the CO field that are waiting to be explored and answered. 

## Figures and Tables

**Figure 1 ijms-23-03573-f001:**
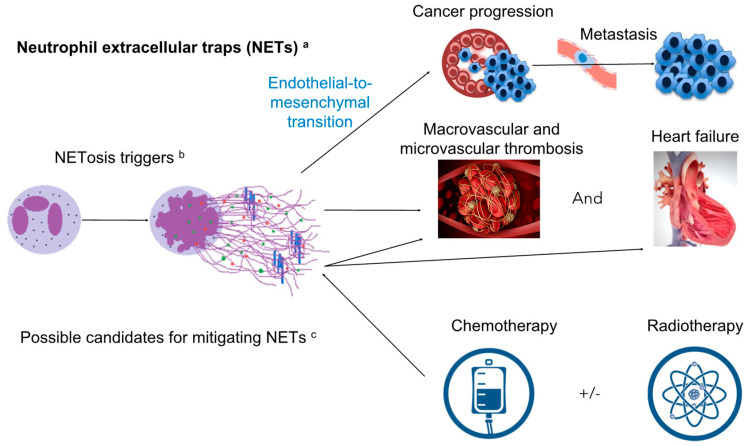
Summary of potential roles of neutrophil extracellular traps (NETs) in cancer treatment-related cardiovascular toxicity. Neutrophil extracellular traps (NETs) ^a^, bearing DNA strands and released proteins containing myeloperoxidase, Cathepsin G, neutrophil elastase (NE), and proteinase 3 (PR3), a series of phospholipases, hypochlorous acid (HOCl) and pro-IL (interleukin)-1α; NETosis triggers ^b^, including activated platelets while in contact with collagen, IL-8, TNF, pathogens, PMA…etc. Possible candidates for mitigating NETs burden to increase cancer survival and reduce cancer treatment-related cardiovascular toxicity (CTRCT) ^c^: including (1) Potent anti-platelet therapy, (2) Deoxyribonuclease (DNase I), (3) Inhibition of peptidyl arginine deiminase-4 (PAD-4), (4) Inhibition of myeloperoxidase (MPO), (5) Anti-cytokine therapy (e.g., targeting IL-1 isoforms, IL-8, or IL-6), (6) Inhibition serine proteases.

**Figure 2 ijms-23-03573-f002:**
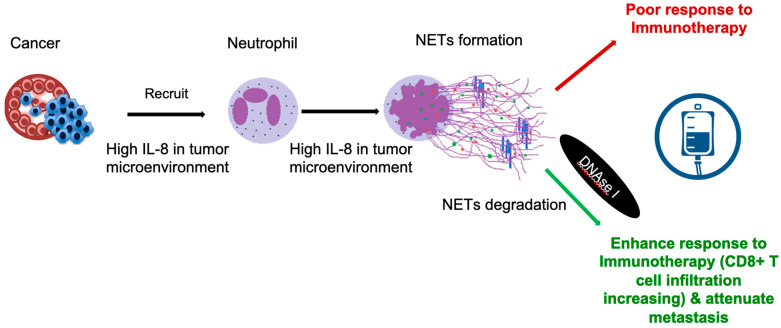
The high IL-8 in tumor environment recruits neutrophil and enhances NETs formation, which causes resistance to immunotherapy. DNase I reverses this situation.

**Table 1 ijms-23-03573-t001:** Unsolved questions and future prospective studies in Cardio-Oncology.

**Prediction of CTRCD**	1. Compare Individual NETs with current standard -GLS and hsTn 2. Combination with NETs, GLS and hsTn better than GLS and hsTn?
**Prediction of thrombosis Events including VTE, ACS and stroke**	1. Compare circulating components of NETs with Khorana score 2. New score system with NETs to predict thrombosis events
**NETs with mLDL & Atherosclerosis after Cancer Treatment**	1. The causality between NETs and mLDL formation after cancer treatment 2.NETs burden after cancer treatment with atherosclerosis development
**NETs Clearance with Cancer Recurrence, Metastasis and CTRCT**	1. NETs clearance to prevent cancer recurrence, metastasis, and CTRCT 2. The best solution/mechanism for NETs clearance

NETs: neutrophil extracellular traps; CTRCD: cancer therapeutics-related cardiovascular dysfunction; GLS: global longitudinal strain; hsTn: high-sensitivity cardiac troponin; VTE: venous thromboembolism; ACS: acute coronary syndrome; Khorana score: a score to predict cancer VTE; mLDL: modified low-density lipoprotein; CTRCT: cancer treatment-related cardiovascular toxicity.

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
