# Peer review of "Potential Role of Neutrophil Extracellular Traps in Cardio-Oncology"

_ijms, 2022, doi:10.3390/ijms23073573_

Round 1

Reviewer 1 Report

Cheng et al., review is of crucial importance for the cardio-oncology field and more particularly to cancer research. All along the manuscript the authors describe in a very well written manner the importance of Neutophils and Neutrophil Extracellular Traps (NETs) cancer treatment related cardio-vascular toxicity.

Although the manuscript is very well written and easy to understand some improvements are required before publication:

As this is a review about NETs and Cardio-oncology one won’t see the connection with COVID-19 disease ; although cardiovascular patients present higher risk factors.

Author Response

Point-by-point response :

Thank you very much for your very positive comments on our manuscript. We have deleted the paragraph- “NET with respect to Coronavirus disease 2019 (COVID-19) disease.” A new author also helped English editing in this version.

Reviewer 2 Report

The significant proportion on cancer survivors suffer from cardiovascular complications due to treatment-related cardiovascular toxicity, including the influence of the therapy itself and immunological changes induced by treatment. Particularly, neutrophil extracellular traps-induced thrombosis and microvascular thrombosis, direct toxicity and disregulation of inflammatory responses in myocardium and vessel walls may significantly contribute to the pathogenesis of cardiovascular complications in cancer. Although the association of NETs with thrombosis in cancer is well investigated in the literature previously, the aspects of possible cardiotoxicity of NETs as well as the changes in NET formation induced by different anti-cancer agents represent the topic of high importance and make the review interesting for the readers.

Nevertheless, several points should be addressed to improve the quality of the manuscript.

I. I would suggest to better structurize the manuscript, as several related to each other parts are localized in chapters far from each other in the text, separated with general information (eg cardiotoxicity due to chemo- and rediotherapy, then NETs in cancer, immunotherapy, then NETs in cardiovascular diseases).

One of the possible solutions could be to use the following order:
1. Intro to the cardio-oncology.
2. NETs (definition, components, mechanism of toxicity with the focus on cardiovascular toxicity and contribution to the cardiovascular events, including heart failure  doi: 10.1155/2021/6687096 , doi: 10.1161/JAHA.121.023800). 
3. NETs in cancer itself.
4. Effects of the different treatments on neutrophil amount, accumulation in tissues, activation and possible effects on NET formation (different chemotherapy agents like doxorubicin, cyclophosphamide, methotrexate, bleomycin, vinblastine, cisplatin; irradiation; monoclonal antibodies and other immunological, eg anti-PD1)
For each treatment modality, I would suggest to include the data (when possible) of in vitro experiments, animal models, clinical studies and speculate if and how the changes in neutrophil properties could impact NET formation and heart and vessel damage.
5. Possible treatment modalities.
6. Unsolved questions.

The following literature could be included: 
PMID: 34659877, doi: 10.21037/tlcr-21-152 , https://doi.org/10.1038/s41598-020-63796-w , DOI: 10.1165/rcmb.2019-0433OC , doi: 10.1016/j.vetimm.2020.110011 , doi: 10.3390/cancers11091328 , DOI: 10.1200/JCO.2018.36.15_suppl.e12579 , DOI: 10.1111/j.1538-7836.2011.04502.x ,  https://doi.org/10.3889/oamjms.2020 , doi: 10.1073/pnas.1613187113

II. On my opinion, chapter about COVID-19 is not in scope of the manuscript and might be removed.

III. Please reduce the text in the figures, in order to make them more demonstrative and understandable from the first view.

IV. The text will benefit from the proof-reading by a native speaker to correct the multiple mistakes (eg, in abstract line 13 "proteinases", 13-14 "play an important role in", 15 "shown to be" etc) and style.

Author Response

Point-by-point responses: 

Point (I): Thank you very much for agreeing that NETs represent the topic of high importance and make this review interesting for the readers. We have reorganized the paper as you suggested. We sincerely appreciate you for suggesting these essential references in NETs studies. These important references have been all included in our review paper now.

Point (II): We agreed with you that COVID-19 is not in scope of this manuscript. We have removed the “NET with respect to Coronavirus disease 2019 (COVID-19) disease.”

Point (III): Thanks for your suggestions. We have deleted some texts and also enlarged the text font in the figures.

Point (IV): This manuscript has been already revised by a native speaker.